# Dynamics and consequences of the HTLV-1 proviral plus-strand burst

Saumya Ramanayake[1], Dale A. Moulding[2], Yuetsu Tanaka[3], Abhyudai Singh[4], Charles R. M. Bangham[1]*

**1** Department of Infectious Diseases, Faculty of Medicine, Imperial College London, London, United Kingdom, **2** Light Microscopy Core Facility, Great Ormond Street Institute of Child Health, University College London, London, United Kingdom, **3** Department of Infectious Disease and Immunology, Okinawa-Asia Research Center of Medical Science, Faculty of Medicine, University of the Ryukyus, Nishihara, Okinawa, Japan, **4** Department of Electrical and Computer Engineering, University of Delaware, Newark, Delaware, United States of America

* c.bangham@imperial.ac.uk

**Data Availability Statement:** All relevant data are within the manuscript and its Supporting Information files.

**Funding:** This work was supported by the Wellcome Trust UK (Investigator Awards 100291

## Abstract

Expression of the transcriptional transactivator protein Tax, encoded on the proviral plus-strand of human T-cell leukaemia virus type 1 (HTLV-1), is crucial for the replication of the virus, but Tax-expressing cells are rarely detected in fresh blood *ex vivo*. The dynamics and consequences of the proviral plus-strand transcriptional burst remain insufficiently characterised. We combined time-lapse live-cell imaging, single-cell tracking and mathematical modelling to study the dynamics of Tax expression at single-cell resolution in two naturally-infected, non-malignant T-cell clones transduced with a short-lived enhanced green fluorescent protein (d2EGFP) Tax reporter system. Five different patterns of Tax expression were observed during the 30-hour observation period; the distribution of these patterns differed between the two clones. The mean duration of Tax expression in the two clones was 94 and 417 hours respectively, estimated from mathematical modelling of the experimental data. Tax expression was associated with a transient slowing in cell-cycle progression and proliferation, increased apoptosis, and enhanced activation of the DNA damage response pathways. Longer-term follow-up (14 days) revealed an increase in the proportion of proliferating cells and a decrease in the fraction of apoptotic cells as the cells ceased Tax expression, resulting in a greater net expansion of the initially Tax-positive population. Time-lapse live-cell imaging showed enhanced cell-to-cell adhesion among Tax-expressing cells, and decreased cell motility of Tax-expressing cells at the single-cell level. The results demonstrate the within-clone and between-clone heterogeneity in the dynamics and patterns of HTLV-1 plus-strand transcriptional bursts and the balance of positive and negative consequences of the burst for the host cell.

## Author summary

Human T-cell leukaemia virus type 1 (HTLV-1) causes disabling or fatal diseases in up to 10% of the infected individuals. The expression of viral protein Tax is essential to cause

and 207477 to CRMB) and the Medical Research Council UK (Project Grant MR/K019090/1 to CRMB). The funders had no role in study design, data collection and analysis, decision to publish, or manuscript preparation.

**Competing interests:** The authors have declared that no competing interests exist.

new infections and contributes to HTLV-1-associated diseases. The proviral plus-strand, which encodes Tax, is expressed in intense intermittent bursts. However, the kinetics of Tax expression and its short and longer-term impact on the infected cell are not well understood. We combined live-cell imaging and mathematical modelling to study Tax expression kinetics in two naturally-infected, non-malignant T-cell clones. Single-cell analysis showed five patterns of Tax expression, with most Tax-positive cells expressing continuously during the 30-hour imaging. The average duration of Tax expression in the two clones was 94 and 417 hours respectively, by mathematical modelling. Tax expression correlated with transiently decreased proliferation, increased apoptosis, enhanced activation of DNA damage response pathways and delayed progression through the cell-cycle. Extended observation showed an increase in the proportion of proliferating cells and a decrease in the percentage of apoptotic cells as cells ceased Tax expression, resulting in a greater net growth of the originally Tax-positive population. Tax-expressing cells also formed cell clumps and showed reduced cell movement. These results help to reconcile the previous apparently conflicting observations of the impact of Tax on the host cell.

## Introduction

Human T-cell leukaemia virus type 1 (HTLV-1), the first isolated human retrovirus, causes an infection in approximately 10 million people worldwide [1]. Around 5–10% of the infected individuals develop HTLV-1-associated diseases, usually after a long incubation period lasting several years to decades [2]. The primary HTLV-1-associated diseases, adult T-cell leukaemia/lymphoma (ATL) and HTLV-1-associated myelopathy/tropical spastic paraparesis (HAM/TSP), still lack satisfactory treatments.

HTLV-1 encodes structural, enzymatic, regulatory and accessory proteins from its nine kilobase genome through ribosomal frameshifting, bidirectional transcription and alternative splicing [3]. In addition to the exogenous retroviral structural and enzymatic proteins Gag, Pro, Pol and Env, the virus also encodes several regulatory and accessory proteins in the pX region of the viral genome. The Tax protein (expressed from the plus-strand of the provirus) and the HBZ protein (expressed from the minus-strand) are the most studied among regulatory HTLV-1 proteins and have many opposing functions [4]. Concerted expression of these two proteins is crucial for the persistence and pathogenesis of HTLV-1 provirus.

The regulation of HTLV-1 expression *in vivo* remains poorly characterised. HTLV-1 virions and viral products are rarely detected in freshly isolated infected blood [5, 6]. However, a chronically activated CTL response against HTLV-1 antigens indicates persistent proviral expression at the cell population-level *in vivo* [7]. Recent *in vitro* studies showed that both plus- and minus-strands of HTLV-1 are expressed in intermittent bursts [8–10].

Tax protein binds or modulates the activity of many host proteins [11] and alters the transcription of many host genes [12]. However, the consequences of HTLV-1 proviral expression on the behaviour of the infected cell are less well understood. Several studies have reported opposite Tax-associated effects, for example both proliferation and apoptosis [13–16]. These apparently contradictory effects of Tax are likely due to variation in the level and duration of Tax expression and the different cell models used. The impact of the plus-strand burst on cellular behaviour, including cell motility and cell-to-cell adhesion, is not sufficiently understood. Characterising the dynamic pattern of the plus-strand burst and its consequences in a physiologically relevant model is crucial for a comprehensive understanding of the persistence and pathogenesis of HTLV-1 infection.

In this study, we used Tax protein as a marker of the HTLV-1 proviral plus-strand transcriptional burst. Using time-lapse live-cell imaging followed by single-cell tracking of naturally HTLV-1-infected CD4$^+$ T-cell clones, we show heterogeneity in Tax expression both between cells and between individual infected subjects, and that Tax expression in each cell typically lasts for over 30 hours. We found that Tax expression in naturally-infected T-cell clones correlated with decreased proliferation, enhanced activation of the DNA damage response (DDR) pathways, slower cell-cycle progression and increased apoptosis. However, longer-term follow-up of Tax-expressing cells revealed that the plus-strand burst is followed by decreased apoptosis and increased proliferation, thus compensating for the detrimental immediate effects of the Tax bursts. Finally, using time-lapse live-cell imaging, we show the association of Tax expression with cell-to-cell adhesion, and with decreased cell motility at the single-cell level. We combined cell population-level and single-cell analysis of naturally HTLV-1-infected T-cell clones to identify and quantify the heterogeneity and dynamics of Tax expression and the consequent effects of the plus-strand transcriptional burst on the behaviour of the infected T-cell.

## Results

### Long Tax bursts in naturally HTLV-1-infected, non-malignant T-cell clones

We have previously shown, using single-molecule RNA fluorescent *in situ* hybridisation (smFISH), that both sense- and antisense-strands of HTLV-1 in *ex vivo* PBMCs and naturally-infected T-cell clones are expressed in intermittent transcriptional bursts [8, 9]. Although smFISH is a powerful technique to quantify transcriptional bursting at single-cell resolution, it requires fixation of cells, leading to the loss of temporal information. The dynamic pattern of expression of the HTLV-1 proviral plus-strand, as indicated by the presence of the transcriptional transactivator protein Tax, is likely to be a significant determinant of the persistence and pathogenesis of the virus, promoting selective proliferation of the infected cell while minimising exposure to immune-mediated selection.

To study the dynamics of Tax expression, we generated a Tax reporter system by combining a chimeric HTLV-1 promoter containing 18 copies of the Tax-responsive element (TRE) [17] with a short half-life version of enhanced green fluorescent protein (d2EGFP) (Fig 1A).

We used stable transduction by lentiviral particles to express the Tax reporter construct in two naturally-infected T-cell clones (TBX4B and TBW 11.50) isolated from two different individuals, each clone carrying a single copy of the HTLV-1 provirus. Fluorescence-activated cell sorting (FACS) followed by intracellular staining with an anti-Tax monoclonal antibody confirmed a strong association between Tax and d2EGFP expression (Fig 1B and 1C). A cyclohex-imide chase assay confirmed the short half-life of d2EGFP (3.39 hours; 95% confidence interval, 2.81–4.29 hours) (S1 Fig). After validating the Tax reporter system in the two clones, we performed live-cell imaging followed by cell tracking to study the dynamics and heterogeneity of Tax expression at single-cell resolution. Live-cell imaging revealed five different patterns of Tax expression during the observation period of 30 hours in both clones (Fig 1D). We designate the different patterns as follows: continuous, reactivating, silencing, fluctuating and transient (Fig 1E). Although the distribution of Tax expression patterns differed between the two clones, continuous expression predominated in both clones (64% and 46% of Tax-expressing cells in d2EGFP TBX4B and d2EGFP TBW 11.50, respectively) (Fig 1D and S1 Video). Live-cell imaging of FACS-sorted non-Tax-expressing cells showed a higher proportion of spontaneously reactivated cells among d2EGFP TBW 11.50 compared to d2EGFP TBX4B (S2 Fig).

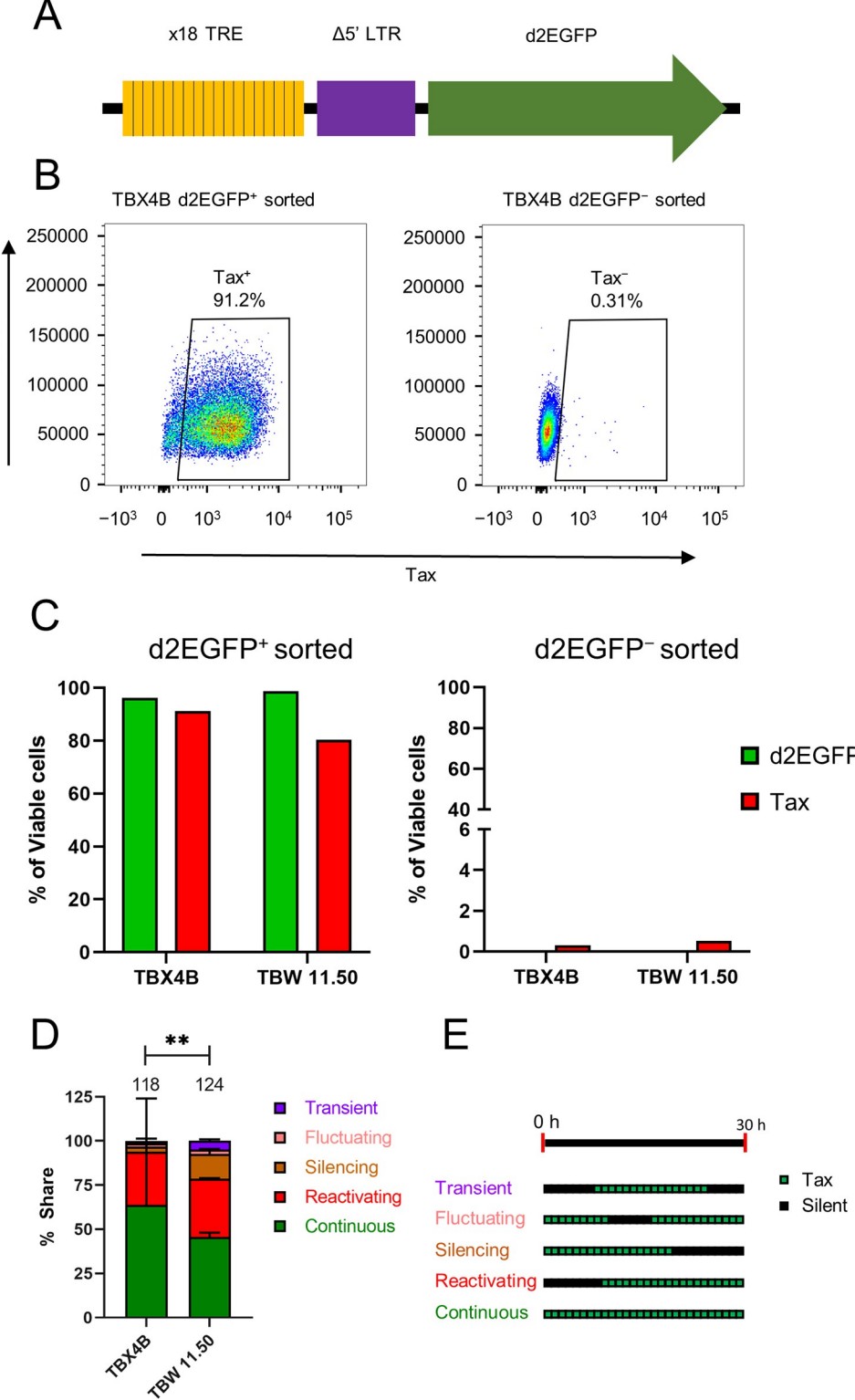

**Fig 1. Live-cell imaging reveals multiple patterns of Tax expression in naturally-infected T-cell clones.** (A) Tax-reporter construct, containing 18 copies of the Tax-responsive element and a truncated HTLV-1 LTR upstream of the *d2EGFP* gene. (B) Representative flow cytometric plots from clone d2EGFP TBX4B depicting Tax protein expression in FACS-sorted d2EGFP⁺ and d2EGFP⁻ cells. (C) d2EGFP and Tax protein expression among FACS-sorted d2EGFP⁺ and d2EGFP⁻ cells from clones d2EGFP TBX4B and d2EGFP TBW 11.50 from a single experiment. (D) Time-lapse

imaging was performed at 20-minute intervals for 30 hours on d2EGFP TBX4B and d2EGFP TBW 11.50 clones. Individual d2EGFP$^+$ cells were identified and tracked to determine the patterns of d2GEFP (Tax) expression at single-cell resolution over the 30-hour imaging period. Data shown are the mean and SEM from two independent experiments. The numbers at the top of the bars denote the total number of individual cells analysed for each clone from two independent experiments. (E) Five patterns of Tax expression were observed during 30 hours of observation, revealed by live-cell imaging: continuous–d2EGFP-positive throughout the period of observation; reactivating–starting d2EGFP expression during the observation period; silencing–ceasing d2EGFP expression during the imaging period; fluctuating–when a d2EGFP-positive cell became d2EGFP-negative and returned to d2EGFP positivity; and transient–short-lived period of d2EGFP expression during the period of observation. Chi-squared test was used to analyse the data in panel D. $^{**}$ P $<$ 0.01.

We used our experimental data in a random telegraph model (see S1 Text) to estimate the average Tax burst duration. The average duration of Tax expression was 94.2 hours (95% confidence interval, 64.4–157.7 hours) and 416.8 hours (95% confidence interval, 218.7–2782.0 hours) for d2EGFP TBW 11.50 and d2EGFP TBX4B, respectively.

## Consequences of the Tax expression for the host cell

We recently reported changes in host transcription during the successive stages of a Tax burst in naturally HTLV-1-infected T-cell clones [12]. These changes involved several cellular processes, including proliferation, cell-cycle, DDR and apoptosis. We wished to investigate the impact of the long Tax bursts observed in the two naturally HTLV-1-infected T-cell clones on the cellular processes of the infected T-cells, including proliferation, DNA damage, cell-cycle progression and apoptosis. We quantified the percentage of proliferating cells among Tax-expressing and non-Tax-expressing cells in the two clones, TBX4B and TBW 11.50, by intra-cellular staining with the proliferation marker Ki-67 [18]. The percentage of Ki-67$^+$ cells was significantly higher among non-Tax-expressing cells than in Tax-expressing cells (Fig 2A), suggesting that Tax expression in these naturally-infected clones is associated with decreased cell proliferation in the short term.

There is evidence that Tax can cause DNA damage by impairing the DNA damage repair process and inducing genotoxic mediators [19–21]. However, most of this evidence comes from recombinant plasmid-based expression systems in cells that are not the natural host cell type, or Tax-immortalised cell lines, or HTLV-1-infected transformed cell lines in which Tax is typically expressed at high and sustained levels, unlike naturally-infected CD4$^+$ T-cells. Phosphorylation of the variant histone H2AX at serine 139 to form γ-H2AX is an indicator of the activation of DDR pathways [22]. We investigated Tax-associated DNA damage by intra-cellular co-staining of Tax and γ-H2AX in clones TBX4B and TBW 11.50. The percentage of γ-H2AX positivity was significantly higher among Tax$^+$ cells than in Tax$^-$ cells, suggesting that a higher portion of Tax-expressing cells had incurred DNA damage than non-Tax-expressing cells (Fig 2B).

There is evidence that Tax can induce ROS-mediated DNA damage in primary human CD4$^+$ T-cells and transformed cell lines transduced with viral particles expressing Tax [15, 21]. We therefore investigated if Tax-associated ROS production could explain the enhanced DNA damage observed in Tax-expressing cells. We investigated intracellular ROS production in d2EGFP (Tax)-expressing and non-d2EGFP-expressing cells by flow cytometric analysis using the non-fluorescent CellROX Deep Red probe, which fluoresces when oxidised by ROS [23]. The cellular oxidative stress inducer tert-butyl hydroperoxide (TBHP) induced ROS in the two HTLV-1-infected T-cell clones, d2EGFP TBX4B and d2EGFP TBW 11.50 (S3 Fig Panel A). There was no difference between d2EGFP$^+$ (Tax$^+$) and d2EGFP$^-$ (Tax$^-$) cells in the proportion of ROS-positive cells (S3 Fig Panel B). We conclude that ROS did not mediate the observed increase in DNA damage in Tax-expressing T-cell clones.

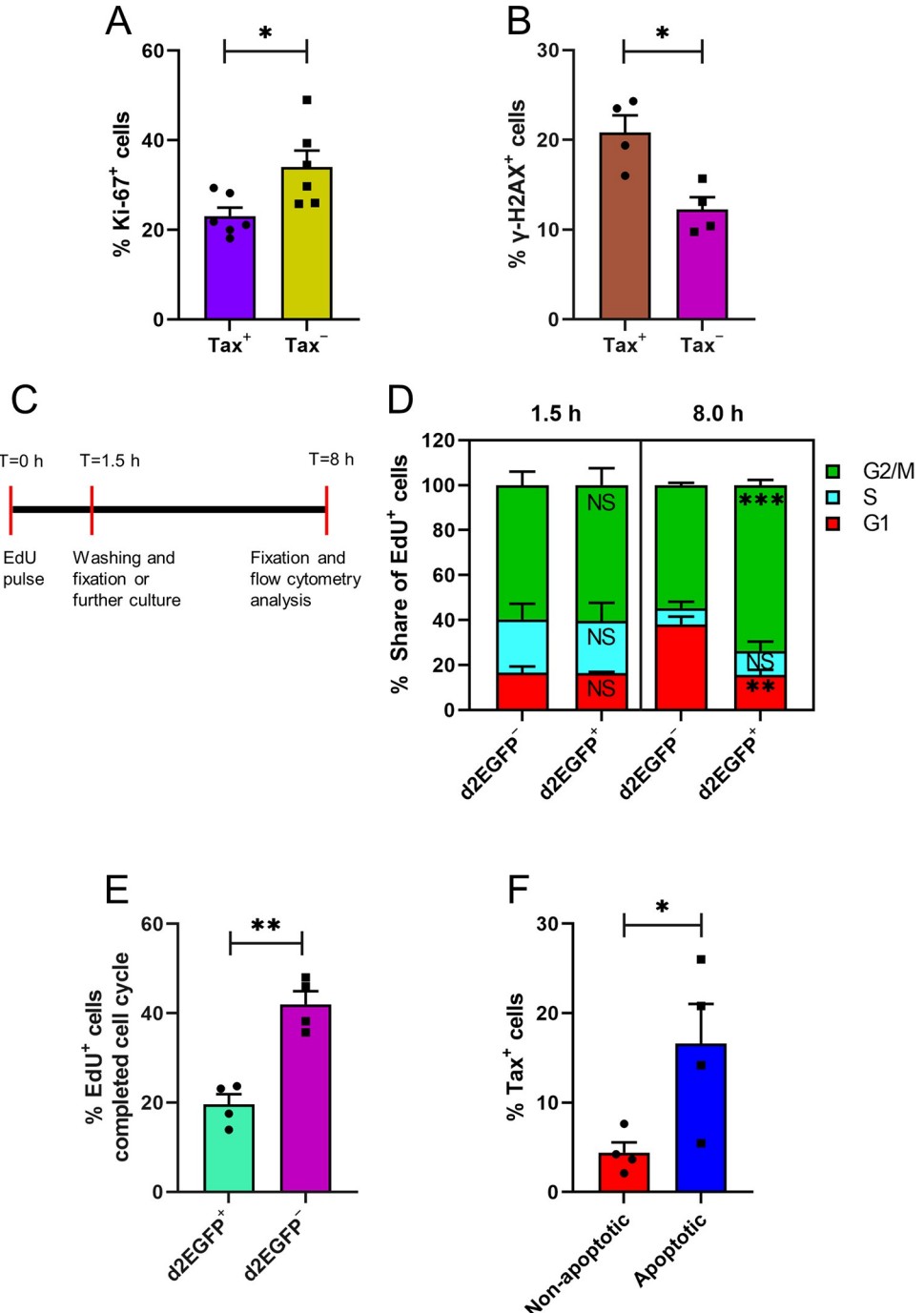

**Fig 2. Short-term impact of Tax expression on the host cell.** (A and B) Flow cytometric analysis of Ki-67 and γ-H2AX in Tax⁺ and Tax⁻ cells from clones TBX4B and TBW 11.50. (C) Experimental design of the EdU pulse-chase assay to determine the cell-cycle progression. d2EGFP TBX4B and d2EGFP TBW 11.50 cells were pulsed with EdU for 1.5 h. After washing, half of the cells were fixed, and the other half were cultured for further 6.5 hours before fixation and flow cytometric analysis. (D) Distribution of EdU-labelled d2EGFP⁺ and d2EGFP⁻ cells in different phases of the cell-cycle at 1.5 and 8 hours. (E) Percentage of EdU-labelled cells among d2EGFP⁺ and d2EGFP⁻ that had completed the previous cell-cycle to enter G1 of the next cell-cycle. (F) Tax expression level in viable apoptotic and non-apoptotic cells from clones TBX4B and TBW 11.50. Bar plots represent the mean and SEM from two to three independent experiments using two T-cell clones. * $P < 0.05$, ** $P < 0.01$, *** $P < 0.001$ (unpaired Student's t-test).

A hallmark of the DDR is the activation of cell-cycle checkpoints that delay progression through the cell-cycle, allowing the damage to be repaired [24]. Since Tax expression was associated with increased DNA damage, we investigated whether this damage resulted in impaired cell-cycle progression. We used a previously described flow cytometric assay that combines DNA content labelling (FxCycle Violet) and the cellular uptake of a thymidine analogue, 5-ethynyl-2′-deoxyuridine (EdU), to analyse cell-cycle progression of d2EGFP (Tax)-expressing and non-d2EGFP-expressing cells in d2EGFP TBX4B and d2EGFP TBW 11.50 clones [10]. We pulsed d2EGFP TBX4B and d2EGFP TBW 11.50 cells with EdU for 1.5 hours (Fig 2C). Following the EdU pulse, the cells were washed, and half of the cells were fixed immediately, while the other half was cultured for further 6.5 hours before fixation and flow cytometric acquisition. The distribution of EdU-positive cells in the cell-cycle phases at the end of the 1.5-hour pulse period was similar in the d2EGFP (Tax)-positive and d2EGFP-negative cells (Figs 2D and S4). However, after 8 hours, the cell-cycle phase distributions of the two populations were markedly different, with most EdU-tagged d2EGFP-positive cells accumulating in G2/M (Fig 2D). Twice as many EdU-labelled d2EGFP-negative cells had progressed to G1 of the next cell-cycle, compared to the EdU-labelled d2EGFP-positive cells (Figs 2E and S5). These results show that Tax expression in these clones was associated with a slower progression through G2/M. EdU uptake is a marker of T-cell proliferation [25]. In accordance with the Ki-67 staining results, we observed reduced levels of EdU uptake at the end of the 1.5-hour pulse period in d2EGFP-positive cells compared to d2EGFP-negative cells (S6 Fig).

A significant consequence of irreparable DNA damage is apoptosis [26]. Having observed Tax-associated DNA damage, we investigated whether Tax expression is also associated with increased levels of apoptosis. We evaluated Tax expression in FACS-sorted viable, apoptotic (annexin V$^+$) and viable, non-apoptotic (annexin V$^-$) cells from clones TBX4B and TBW 11.50. We found a higher proportion of apoptotic cells expressed Tax than non-apoptotic cells, indicating that Tax expression was associated with an increased risk of apoptosis (Fig 2F).

## The longer-term impact of Tax expression on the host cell

The results described above demonstrated immediate adverse effects associated with Tax expression, including decreased proliferation, increased DNA damage, slower progression through the cell-cycle and increased apoptosis. To study the longer-term impact of Tax expression in naturally-infected T-cell clones, we FACS sorted d2EGFP$^+$ (Tax$^+$) and d2EGFP$^-$ (Tax$^-$) cells from clone d2EGFP TBW 11.50 and followed them for 14 days, quantifying d2EGFP (Tax) expression, expansion (cell number), proliferation (Ki-67), and apoptosis (annexin V) on days 0, 4, 7, 11 and 14 (Fig 3A).

Tax-expressing cells isolated by FACS reached the steady-state during the second week of post-sort culture following a monotonic decline (P = 0.0083), whereas the non-Tax-expressing cells had reached equilibrium by Day 4 and showed no significant trend during the culture period (P = 0.408) (Fig 3B). The percentage of apoptotic cells was greater in the sorted Tax-expressing cells during the first week of culture but fell below the percentage observed in sorted non-Tax-expressing cells during the second week, when Tax expression had ceased in most initially Tax-positive cells (Fig 3C). Annexin V positivity of sorted Tax-negative cells showed a monotonic increase (P = 0.0083), while in sorted Tax-expressing cells, there was no significant trend (P = 0.592) (Fig 3C). The proportion of Ki-67$^+$ cells in sorted Tax$^+$ cells increased as the cells ceased expressing Tax, reaching 4.5-fold higher than the day 0 level at the end of the 14-day culture (Fig 3D). The proportion of Ki-67-positive cells in sorted Tax$^-$ cells fell to 0.9-fold at the end of day four before rising to 2.2-fold higher than the day 0 level at the end of the two-week culture (Fig 3D). The expression of Ki-67 in sorted Tax$^-$ cells showed a

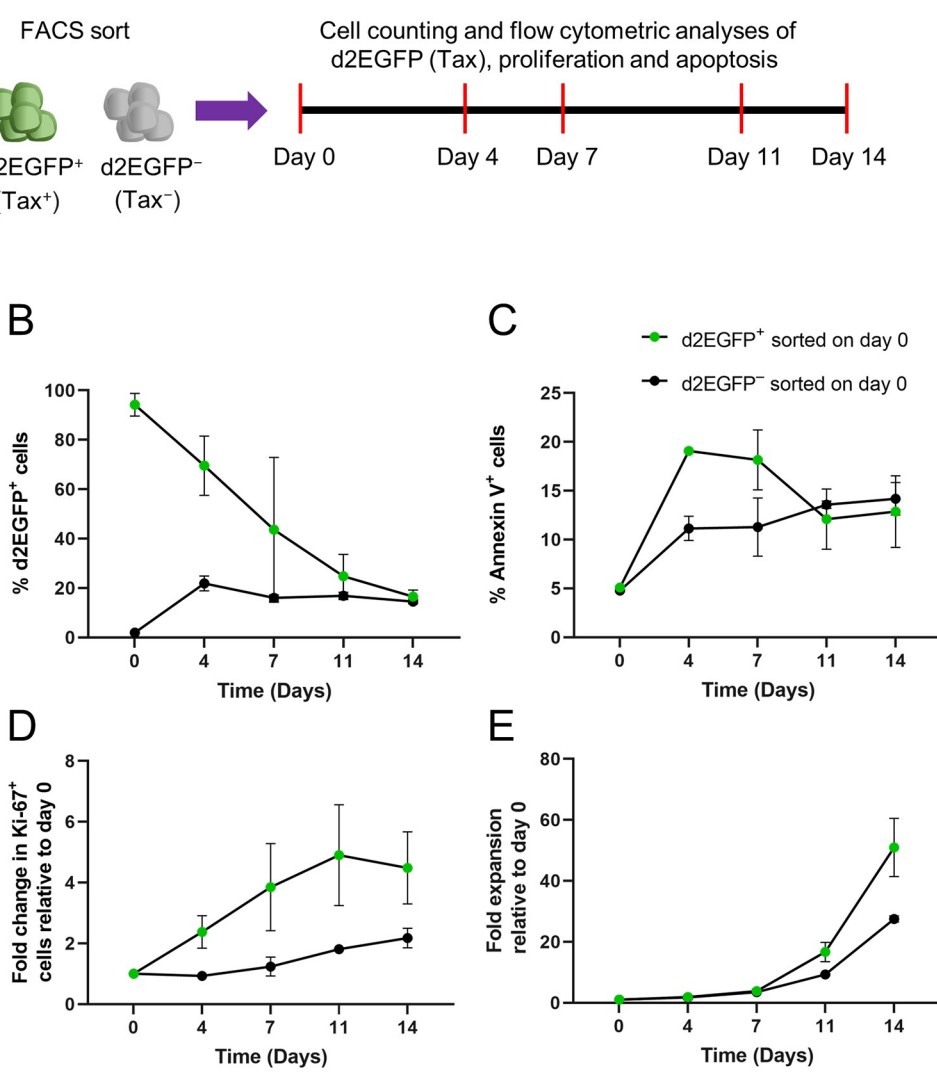

**Fig 3. Longer-term consequences of Tax expression on the host cell.** (A) Experimental design to determine the longer-term impact of Tax expression. d2EGFP$^+$ (Tax$^+$) and d2EGFP$^-$ (Tax$^-$) cells were FACS sorted and cultured for 14 days. Cell counts and flow cytometric analysis of d2EGFP (Tax) expression, proliferation and apoptosis were performed on days 0, 4, 7, 11 and 14 of the culture. (B) d2EGFP and (C) annexin V expression levels of sorted d2EGFP$^+$ and d2EGFP$^-$ cells. (D) Fold change in Ki-67$^+$ cells and (E) net expansion of sorted d2EGFP$^+$ and d2EGFP$^-$ cultures. Data represent the mean ± SEM from two independent experiments using the clone d2EGFP TBW 11.50. The significance of the trend in d2EGFP$^+$ and d2EGFP$^-$ populations was assessed using the Mann-Kendall test.

significant positive trend (P = 0.042), while sorted Tax$^+$ cells showed a significant positive trend during the first 11 days of the culture (P = 0.042) (Fig 3D). The mean net expansion of sorted Tax-expressing cells over the 14-day culture was 51-fold, compared to the 28-fold expansion of sorted non-Tax-expressing cells, with both populations showing a monotonic increase (P = 0.0083) (Fig 3E). These results demonstrate that the immediate detrimental effects of the plus-strand burst are followed by a decrease in the frequency of apoptosis and an increase in proliferation in the surviving cells, leading to a higher net expansion in the sorted Tax$^+$ population than in the sorted Tax$^-$ population.

## Tax-expressing cells form cell clumps

We observed cell clusters (clumps) in our routine cultures of naturally-infected T-cell clones. Tax-induced cell-to-cell adhesion has been previously reported in rodent cell lines and a Tax-inducible transformed cell line [27, 28]. We therefore investigated whether the cell clumps observed in the T-cell clones were associated with Tax expression. We FACS sorted d2EGFP (Tax)-positive and -negative cells from clones d2EGFP TBX4B and d2EGFP TBW 11.50 and performed time-lapse whole-well imaging at 4-hour intervals for 12 hours. The mean object area of the sorted d2EGFP-positive and -negative cultures was similar at the start (0 h) of imaging, but the mean object area of the d2EGFP-positive cultures was significantly larger than that of the d2EGFP-negative cultures at the end (12 h) of the imaging (Fig 4A and 4B).

We also quantified the mean object area of FACS-sorted viable cells from TCX 8.13, a clone carrying a single type 2 defective HTLV-1 provirus incapable of expressing the plus-strand, and an uninfected clone, TBW 13.50. The mean object area of each clone remained unchanged during the period of observation, showing that the clump formation observed in naturally HTLV-1-infected T-cell clones depended on HTLV-1 plus-strand expression (S7 Fig).

We observed significantly higher ICAM-1 expression among d2EGFP$^+$ cells than in d2EGFP$^-$ cells (S7 Fig). Cell-to-cell contact is crucial for HTLV-1 transmission, through the formation of the virological synapse [29]. Interaction of ICAM-1 on the infected cell with its receptor LFA-1 on the target cell is essential for the cytoskeletal polarisation observed in the virological synapse [30]. We hypothesised that an increase in the interaction between ICAM-1 and LFA-1 could explain the enhanced cell-to-cell adhesion (i.e. clumping) observed in Tax-expressing cells. We used both pharmacological and non-pharmacological interventions to inhibit the LFA-1-ICAM-1 interaction. Two pharmacological agents were used: an ICAM-1 expression inhibitor, A205804 [31] and an allosteric inhibitor of ICAM-1-LFA-1 interaction, A286982 [32]. The two non-pharmacological agents were a monoclonal antibody against a functional epitope of ICAM-1 [33], and a cyclic peptide, cLAB.L, that interacts with the D1 domain of ICAM-1 to inhibit ICAM-1-LFA-1 engagement [34]. DMSO was used as the vehicle control for pharmacological agents, while an isotype control antibody and the non-specific cyclic peptide, RD-LBEC, served as the negative control for anti-ICAM-1 antibody and cLAB.L cyclic peptide, respectively. FACS-sorted d2EGFP$^+$ cells were pre-treated with each respective inhibitor or control for one hour before capturing time-lapse whole-well live-cell images every 4 hours for 12 hours. FACS-sorted d2EGFP$^-$ cells were used as the negative control for cell-to-cell adhesion. The mean object area did not differ between the cultures treated with inhibitors of the LFA-1-ICAM-1 interaction and their negative controls (Fig 4C). The mean object area of the sorted d2EGFP$^-$ cultures was substantially lower than the sorted d2EGFP$^+$ cultures under all treatment conditions. We conclude that LFA-1-ICAM-1 engagement alone does not explain the cell-to-cell adhesion observed in Tax-expressing cells.

Flow cytometric analysis also revealed a significantly higher percentage of CCR4$^+$ cells among d2EGFP$^+$ cells than among d2EGFP$^-$ cells (S8 Fig). It has been shown that Tax-expressing cell lines formed cell-to-cell adhesions with uninfected CCR4$^+$ cells in a CCL22-dependent manner [35]. We also observed higher levels of *CCL22* transcripts among FACS-sorted d2EGFP$^+$ cells than in the d2EGFP$^-$ cells (S9 Fig Panel C). To test the hypothesis that interactions between Tax-expressing CCR4$^+$ and CCL22-producing cells are responsible for the cell clumps observed, we tested the effect of CCR4 inhibition on the cell-to-cell adhesion of Tax-expressing cells. FACS-sorted d2EGFP$^+$ cells were pre-treated with either exogenous CCL22, or the CCR4 antagonist C021, or both, for one hour prior to time-lapse live-cell imaging for 12 hours at 4-hour intervals. The mean object area did not differ between the different treatment conditions at the end of the 12-hour culture and was substantially greater than the

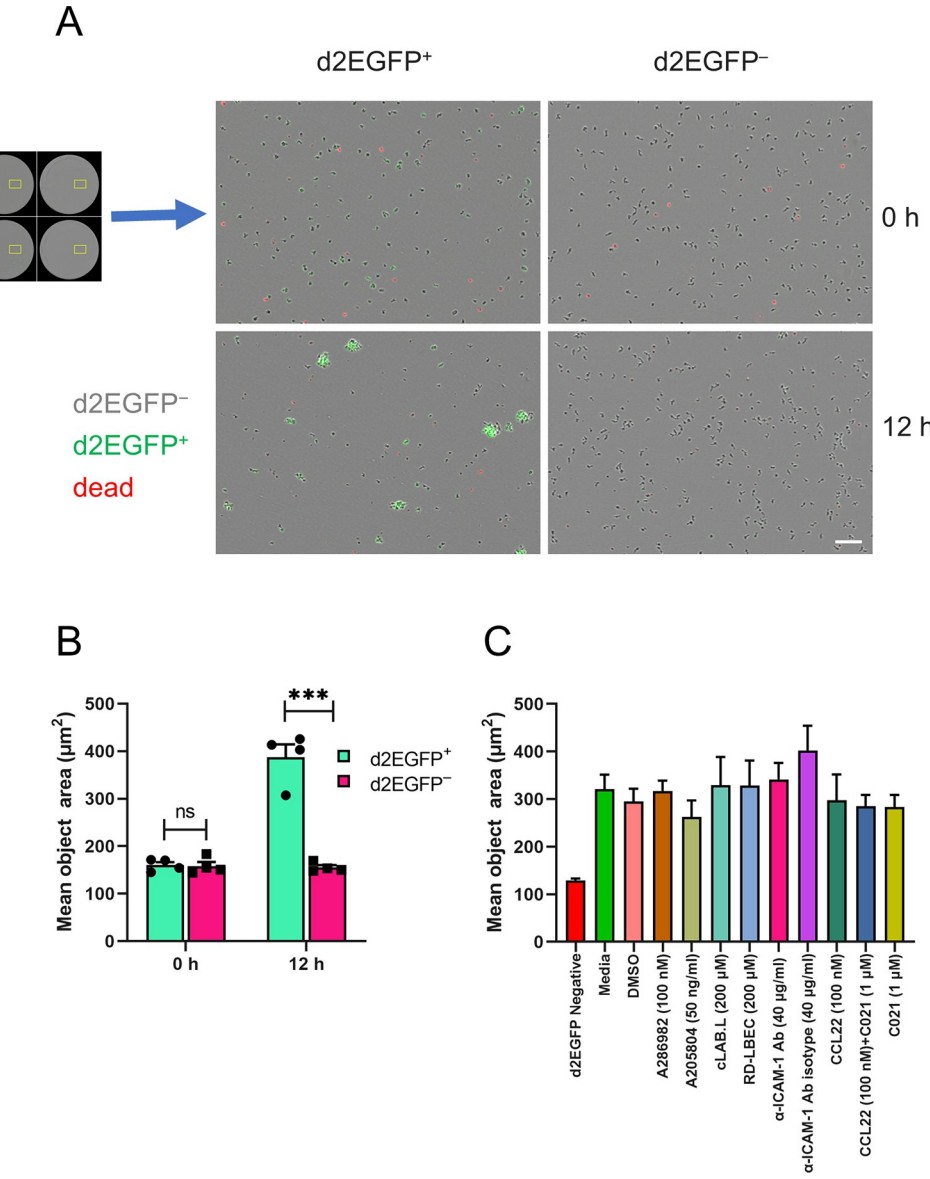

**Fig 4. Tax-expressing cells form cell clumps.** (A) Time-lapse live-cell imaging was performed on FACS-sorted d2EGFP+ (Tax+) and d2EGFP− (Tax−) cells, acquiring whole-well images in phase (d2EGFP− cells), green (d2EGFP+ cells) and red (YOYO-3 Iodide–dead cells) channels every 4 hours for 12 hours. The mean object area in wells seeded with d2EGFP+ and d2EGFP− cells was quantified as outlined in Materials and methods. Representative images from clone d2EGFP TBX4B showing the cell clumps formed among sorted d2EGFP+ cells at the end of the 12-hour culture. Scale bar– 100 μm. (B). Mean object area of sorted d2EGFP+ and d2EGFP− cultures at the beginning (0 h) and the end (12 h) of the culture. Data depict the mean and SEM from two independent experiments using clones d2EGFP TBX4B and d2EGFP TBW 11.50. (C) The effect of inhibition of LFA-1-ICAM-1 interaction or CCR4 expression on clump formation among Tax-expressing cells. FACS-sorted d2EGFP-positive cells from clone d2EGFP TBW 11.50 were pre-treated with agents for 1 hour at the indicated concentrations. FACS-sorted d2EGFP-negative cells were used as the negative control for clump formation. Whole-well time-lapse images were captured in phase contrast, green and red channels over 12 hours at 4-hour intervals. The mean object area at the end of the 12-hour imaging under each treatment condition is shown. The data depict the mean and SEM from two independent experiments. Data in panel B were analysed by an unpaired Student's t-test. *** P < 0.001, ns–not significant.

mean object area of the sorted d2EGFP⁻ cells, the negative control for cell clumping (Fig 4C). These results imply that CCR4 expression was not responsible for the cell-to-cell adhesion observed in Tax-expressing T-cell clones. We hypothesize that there is significant redundancy in the adhesion molecules that mediate the observed cell clumping.

## Tax expression is associated with reduced cellular motility

T-cell motility is vital for normal T-cell functions and for the initiation, propagation and pathogenesis of HTLV-1 infection. Previous studies have demonstrated that Tax-expressing cells have a greater propensity for migration [36, 37]. However, these studies quantified the migratory potential using cell population-level transwell migration assays. We wished to quantify the motility of Tax-expressing and non-Tax-expressing cells at the single-cell level. Since HTLV-1-expressing cells typically grew in clumps, and non-Tax-expressing cells at any given time greatly outnumbered Tax-expressing cells, we mixed the FACS-sorted d2EGFP⁺ (Tax⁺) and d2EGFP⁻ (Tax⁻) cells 1:1 to quantify cell motility. We seeded the cells on to a silicon membrane containing an array of microwells, previously used to study T-cell behaviour [38]. One-hour time-lapse live-cell imaging followed by single-cell tracking demonstrated that d2EGFP-positive cells had more confined tracks than d2EGFP-negative cells (Fig 5A and S2 Video).

Quantitative image analysis revealed that the mean speed of d2EGFP⁺ cells was significantly lower than that of d2EGFP⁻ cells (Fig 5B). Directionality, a measure of the linearity of the cell tracks, did not differ between the two groups (Fig 5C). Mean square displacement (MSD–the displacement of each cell from its starting position) was much greater in d2EGFP⁻ cells than in d2EGFP⁺ cells (Fig 5D). We further analysed MSD data (see Materials and methods) to identify the type of motion in naturally HTLV-1-infected T-cell clones. The motility pattern differed between the two groups: the majority of d2EGFP-negative cells showed superdiffusive motion in the culture (Fig 5E). That is, the MSD significantly exceeded the value expected from diffusive motion. By contrast, a substantial proportion of d2EGFP-positive cells showed diffusive motion. These results show that Tax expression in naturally-infected T-cell clones is associated with decreased cell motility.

## Discussion

A key determinant of the fate between HTLV-1 proviral latency and reactivation is the expression of the transactivator protein Tax. By interacting with various host proteins, Tax modulates several cellular functions, contributing to the persistence and pathogenesis of HTLV-1 [11]. The dynamic pattern of Tax expression is a likely factor determining the fate of the infected cell. In this study, we used Tax (reported by d2EGFP expression) as a marker of proviral plus-strand expression. While Tax itself has been shown to exert diverse effects on the host cell, certain effects reported here may be caused by the other products encoded by the proviral plus-strand. Time-lapse live-cell imaging of two naturally HTLV-1-infected T-cell clones stably transduced with the d2EGFP Tax reporter system identified several distinct patterns of Tax expression during the 30-hour observation period, with most Tax-positive cells expressing the protein continuously for the entire 30-hour period of observation (Fig 1D and S1 Video). The average duration of Tax expression based on the mathematical modelling of the experimental data was 94 hours in clone d2EGFP TBW 11.50 and 417 hours in clone d2EGFP TBX4B (S1 Text). These observations contrast with a study that reported live-cell imaging of Tax expression of an ATL-derived cell line, MT-1, in which transient bursts of Tax lasted an average of 19 hours [10]. The shorter burst duration observed in MT-1 cells may be a consequence of malignant transformation; however, simple clone-to-clone variation in mean burst duration–as illustrated by the difference in mean duration between the two clones in the present study–

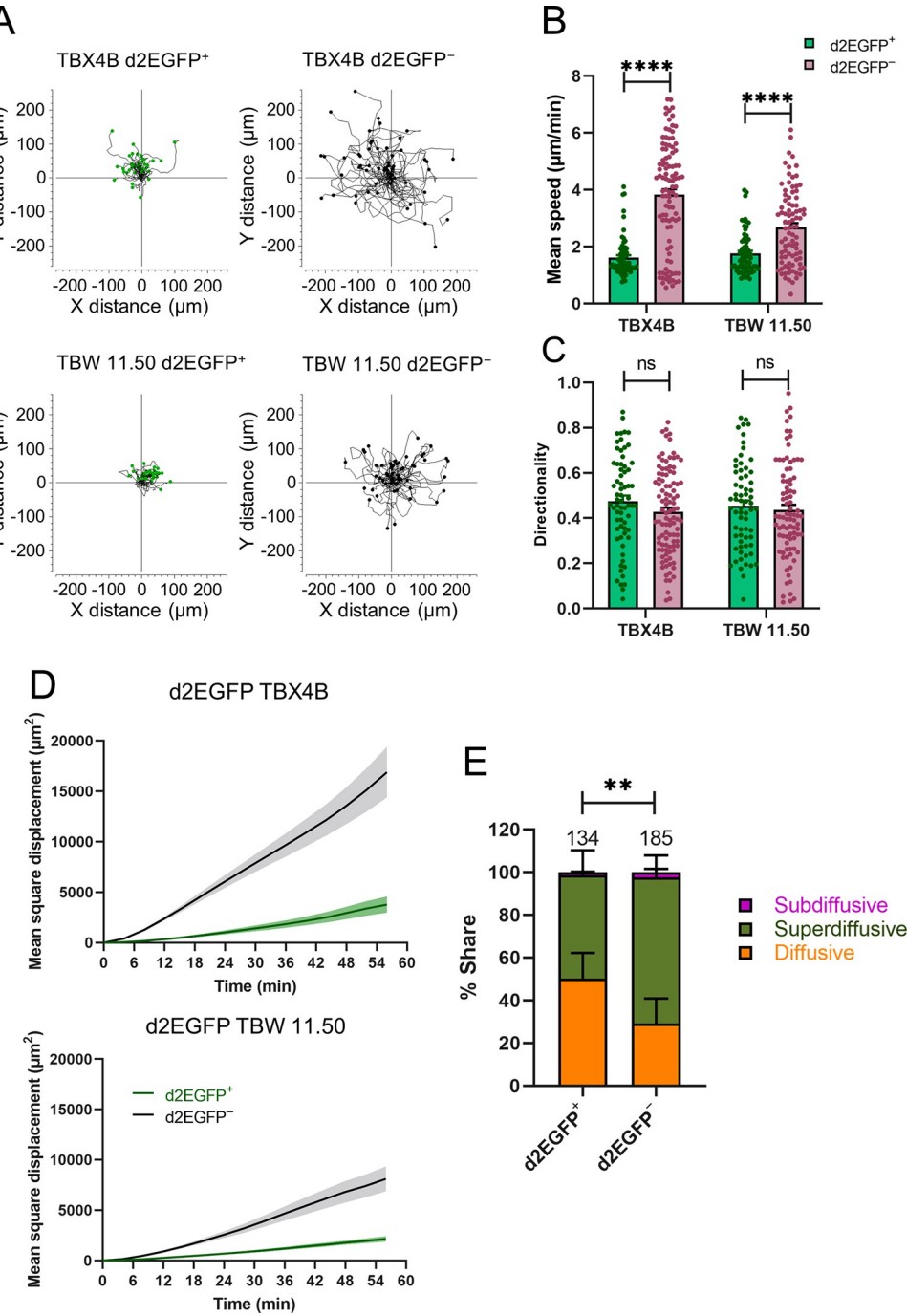

**Fig 5. Tax expression is associated with decreased cellular motility.** (A) FACS-sorted Tax-expressing (d2EGFP$^+$) and non-Tax-expressing (d2EGFP$^-$) cells were mixed 1:1 and seeded into wells containing a polymer insert with an array of 500 μm x 500 μm wells. Time-lapse live-cell imaging was performed for 1 hour, capturing images at either 3- or 4-minute intervals in phase (d2EGFP$^-$ cells), green (d2EGFP$^+$ cells) and red (YOYO-3 Iodide–dead cells) channels. Single-cell tracking and analysis, described in Materials and methods, was used to determine cell motility. Trajectory plots of d2EGFP$^+$ and d2EGFP$^-$ cells from clones d2EGFP TBX4B and d2EGFP TBW 11.50 after transforming the starting coordinates of all cells to a common (0, 0) origin. The trajectories from a single representative experiment of two independent experiments are shown. The number of trajectories shown are: TBX4B d2EGFP$^+$– 42, TBX4B d2EGFP$^-$– 55, TBW 11.50 d2EGFP$^+$– 33, TBW 11.50 d2EGFP$^-$– 56. (B) Mean speed and (C) directionality of d2EGFP$^+$ and d2EGFP$^-$ cells. Number of cells analysed: TBX4B d2EGFP$^+$– 68, TBX4B d2EGFP$^-$– 101, TBW 11.50 d2EGFP$^+$– 67, TBW 11.50 d2EGFP$^-$– 87. The mean and SEM from two independent experiments are shown. (D) Mean square displacement of d2EGFP$^+$ and d2EGFP$^-$ cells from two clones. The mean ± SEM from a single

representative experiment of two independent experiments are shown. Number of cells analysed: TBX4B d2EGFP⁺– 42, TBX4B d2EGFP⁻– 55, TBW 11.50 d2EGFP⁺– 33, TBW 11.50 d2EGFP⁻– 56. (E) Distribution of different motility patterns observed in d2EGFP⁺ and d2EGFP⁻ cells. The mean and SEM from two independent experiments using clones d2EGFP TBX4B and d2EGFP TBW 11.50 are shown. The numbers on top of the bars depict the number of cells analysed. A Mann-Whitney (B and C) or a chi-squared (E) test was used to determine the statistical significance. ** P < 0.01, **** P < 0.0001, ns–not significant.

cannot be excluded. Whereas non-malignant HTLV-1-infected cells typically retain the ability to express Tax, its expression is frequently silenced in ATL clones by several distinct mechanisms [39]. The observed difference in the pattern of Tax expression between the two non-malignant clones in the present study indicates clone-specific factors that regulate proviral expression patterns.

There are several lines of evidence that Tax can induce DNA damage, mainly from studies using HTLV-1-transformed cell lines and cell lines immortalised by Tax [19–21, 40]. Consistent with these reports, a higher proportion of Tax-expressing cells in the naturally-infected clones stained positive for γ-H2AX than non-Tax-expressing cells (Fig 2B). Although γ-H2AX is primarily used as a marker of DNA double-strand breaks (DSBs), γ-H2AX is also formed in response to DNA replication stress [41]. Since Tax expression was associated with less proliferation and lower DNA synthesis (Figs 2A and S6), it is likely that the higher levels of γ-H2AX among Tax-positive cells were due to Tax-associated DSBs. The DDR process activates checkpoints that slow the passage through the cell-cycle and initiate DNA damage repair [42]. There is evidence that Tax can prolong cell-cycle phases [10, 43]. Consistent with these previous reports, Tax expression in the two clones studied here was associated with slower progression through G2/M (Fig 2D). Our live-cell images showed enlarged Tax-expressing cells without undergoing cytokinesis (S1 Video) in accord with a study reporting that Tax-expressing cells became senescent without completing mitosis [43]. We recently reported the changes in expression levels of genes involved in DDR during successive phases of Tax expression [12]. *ATM and Rad3-related (ATR)* mRNA levels followed the pattern of Tax expression, whereas *ataxia-telangiectasia mutated* (*ATM*) mRNA levels fell during the burst. There is also evidence of Tax-induced impairment of ATM activity [20]. These observations suggest that ATR-mediated DDR is the primary repair mechanism of DNA damage induced by Tax in these clones. Irreparable DNA damage leads to either permanent termination of growth (senescence) or programmed cell death (apoptosis) to prevent the aggregation of genetic defects that could lead to malignant transformation. There is evidence of Tax-mediated persistent NF-κB activation leading to cellular senescence [44] through the build-up of R-loops whose processing subsequently induced DSBs resulting in cellular senescence [45]. A significantly greater proportion of apoptotic cells expressed Tax than did non-apoptotic cells (Fig 2F), consistent with the notion that some Tax-expressing cells that had incurred irreparable DNA damage underwent apoptosis. In contrast to a previous report [21], there was no difference in the frequency of ROS-positive cells between Tax-expressing and non-Tax-expressing cells (S3 Fig), suggesting that ROS did not mediate Tax-associated enhanced DNA damage in these naturally HTLV-1-infected T-cell clones.

We have previously shown that reactivation of the HTLV-1 plus-strand resembles an immediate-early stress response gene [46]. In agreement with this observation, long-term follow-up of Tax expression revealed that the Tax-negative population had already reached the steady-state by the fourth day of the post-sort culture, much earlier than the Tax-positive population, which took a longer time, reaching the equilibrium during the second week of the culture (Fig 3A). These differences in the kinetics of spontaneous reactivation and shutdown suggest that the epigenetic modifications that allow reactivation are not quickly restored once

the barrier to reactivation is overcome. An increase in the proportion of proliferative cells, coupled with a decrease in the proportion of apoptotic cells as the cells terminated Tax expression, led to a greater net expansion of the FACS-sorted Tax-positive cultures (Fig 3B, 3C and 3D). The mechanism behind this late post-Tax burst surge of proliferation is unknown. A delayed effect of a host factor upregulated by Tax, or the effect of the viral protein HBZ–which both promotes T-cell proliferation and hinders Tax expression–are potential contributors to this proliferation [47, 48]. The study by Mahgoub et al. showed a Tax-mediated anti-apoptotic effect that extended beyond the Tax-expressing period [10]. A similar phenomenon among Tax-expressing cells that survive the negative effects of long Tax bursts associated with these naturally-infected, non-malignant T-cell clones, leading to a greater net expansion, cannot be ruled out. Another attractive hypothesis is apoptosis-induced compensatory proliferation, a mechanism by which apoptotic cells induce proliferation and survival of surrounding cells for maintaining tissue homeostasis in insects, rodents and human cancers [49–51].

Tax expression is associated with upregulation of the cell adhesion molecule, ICAM-1 [52]. Several studies have demonstrated Tax-induced cell-to-cell adhesion in rodent and HTLV-1 transformed human cell lines [27, 28]. In agreement with these studies, we observed strong cell-to-cell adhesion among Tax-expressing cells, whereas non-Tax expressing cells displayed behaviour similar to an uninfected clone and a clone incapable of expressing the plus-strand (Figs 4A, 4B and S7). Our assay was designed to quantify the clump formation within either Tax-positive or Tax-negative cells but not between Tax-positive and -negative cells. However, live-cell imaging indicated that Tax-expressing cells can contact non-Tax-expressing cells to form cell clumps (see the S2 Video). Unlike HIV-1, cell-free virions of HTLV-1 are rare and weakly infectious, and cell-to-cell transmission through the virological synapse is the main mechanism of HTLV-1 infection [29]. A trigger for the formation of the virological synapse is the engagement of ICAM-1 on the surface of the infected cell with LFA-1 on the surface of the target cell, which causes the polarisation of the infected cell's cytoskeleton towards the cell-to-cell contact [30]. Consistent with previous reports [52, 53], Tax expression was associated with increased ICAM-1 expression frequency (S8 Fig). However, blocking the ICAM-1-LFA-1 interaction did not inhibit the clump formation between Tax-expressing cells, implying that the increase in cell-to-cell adhesion among Tax-expressing cells is independent of the ICAM-1-LFA-1 interaction (Fig 4C). There is evidence of cell-to-cell adhesion between uninfected CCR4+ cells and Tax-expressing cell lines in a CCL22-dependent manner [35]. The frequency of CCR4 expression and the level of *CCL22* transcripts were higher in Tax-expressing cells than in non-Tax-expressing cells (S8 Fig and S9 Fig Panel C). However, pharmacological inhibition of CCR4 did not lead to a reduction in cell-to-cell adhesion among Tax-expressing cells, suggesting that clump formation in Tax-positive cells is independent of interactions between CCR4 and CCL22-expressing cells (Fig 4C). There is evidence from a parallel study by our group that clump formation requires the activation of the phospholipase C gamma 1 (PLCγ1) pathway and is enhanced by the presence of exogenous IL-2 (Prawiro et al., under revision).

T-cells are one of the most motile cell types in the body [54]. There is evidence of the enhanced migratory capacity of HTLV-1-infected cells using cell population-level transwell migration assays [36, 37]. One study showed Tax-mediated upregulation of Gem, a GTPase that facilitates cytoskeleton remodelling to potentiate cell migration [37]. According to a second study, enhanced motility among HTLV-1-infected cells was only partly attributable to Tax, which augmented the levels of the cytoskeleton remodelling phosphoprotein CRMP2 but did not induce its phosphorylation, which is required for CRMP2 function [36]. Single-cell quantification of cell motility using time-lapse live-cell imaging followed by single-cell tracking revealed decreased motility in Tax-expressing cells (Fig 5A, 5B and 5D). The contrast between the observations reported here and the previous studies described above is attributable to the

different experimental design: we quantified random cell motility at single-cell resolution in the 2D plane, whereas the studies mentioned above quantified cell migration through a transwell membrane in response to a chemokine gradient. Tax upregulates several cell adhesion molecules and ligands [52, 53, 55]. We suggest that adhesion of Tax-expressing cells to the polymer surface hindered their motility, resulting in diminished speeds and displacement. Future experiments should be designed to test this hypothesis. The motility pattern differed between Tax-expressing and non-Tax-expressing cells, with most Tax-negative cells showing superdiffusive motility (Fig 5E).

In conclusion, using non-malignant T-cell clones naturally infected with HTLV-1, we showed both within and between clone heterogeneity in the dynamics and patterns of HTLV-1 plus-strand expression. Single-cell analysis and mathematical modelling revealed the long duration of Tax bursts. We observed immediate but transient detrimental effects of long Tax expression, including decreased proliferation, increased apoptosis, enhanced activation of DDR pathways, and slower progression through the cell-cycle. In the longer term, these adverse effects were compensated following the termination of Tax expression, as evidenced by an increase in proliferation and a decrease in the frequency of apoptosis, leading to a higher net expansion. Finally, we showed increased cell-to-cell adhesion and, at the single-cell level, decreased motility associated with Tax expression. These findings highlight the within-clone and between-clone heterogeneity in the kinetics and patterns of HTLV-1 plus-strand expression and the balance between the beneficial and harmful effects of Tax expression on the host cell.

## Materials and methods

### Cell culture

Two naturally-infected T-cell clones harbouring a single copy of HTLV-1 provirus, a T-cell clone naturally-infected with a type 2 defective HTLV-1 provirus and an uninfected T-cell clone were isolated by Cook et al. by limiting dilution of $CD4^+CD25^+$ cells [56]. Two naturally-infected T-cell clones stably expressing the d2EGFP Tax reporter system were generated as described previously [12]. All clones were propagated in complete growth medium containing RPMI-1640 (Sigma-Aldrich) with 20% fetal calf serum (FCS), 50 IU/ml penicillin, 50 μg/ml streptomycin, 2 mM L-glutamine (all from ThermoFisher Scientific) and 100 IU/ml human interleukin 2 (IL-2, Miltenyi Biotec). All HTLV-1 infected T-cell clones were maintained in 10 μM of the integrase inhibitor, raltegravir (Selleck Chemicals), to eliminate subsequent HTLV-1 infections. The two T-cell clones transduced with the d2EGFP Tax reporter system were cultured in 1 μg/ml puromycin dihydrochloride (ThermoFisher Scientific) to avoid the establishment of populations devoid of a functional resistance gene. IL-2 and, where applicable, raltegravir and puromycin dihydrochloride were added at biweekly intervals to the cells cultured in a humidified incubator, maintained at 37˚C, with 5% $CO_2$. The details of the T-cell clones used in this study are given in the S1 Table.

### Flow cytometric analysis of cell surface and intracellular proteins

To evaluate the expression of d2EGFP and cell surface proteins, the cells were washed once in PBS and stained for 5 minutes with 1 μg/ml LIVE/DEAD fixable near-IR dye (ThermoFisher Scientific). The cells were washed once with FACS buffer (PBS supplemented with 5% FCS). If surface protein detection was required, the cells were stained with either anti-ICAM-1-PE (clone HA-58, BioLegend) or anti-CCR4-PE (clone L291H4, BioLegend) or isotype control antibody, IgG1 κ-PE (clone MOPC-21, BioLegend) for 30 minutes. The cells were washed

twice with FACS buffer and fixed with 4% formaldehyde (ThermoFisher Scientific) for 30 minutes. The cells were subsequently washed twice and resuspended in FACS buffer.

To detect intracellular proteins, the cells were first stained for 5 minutes with 1 μg/ml LIVE/DEAD fixable near-IR dye after being washed once with PBS. The cells were washed once with FACS buffer and subsequently fixed for 30 minutes with fixation/permeabilisation buffer of eBioscience Foxp3/transcription factor staining buffer set (ThermoFisher Scientific). They were then washed once with the permeabilisation buffer of the staining buffer set and stained with anti-Tax-AF647 (clone LT-4) and either anti-Ki-67-PE (BioLegend) or anti-γ-H2AX-PE (clone 2F3, BioLegend) in permeabilisation buffer for 30 minutes. IgG3 κ-AF-647 (clone MG3-35, BioLegend) served as the isotype control antibody for the anti-Tax-AF647, and IgG1 κ-PE (clone MOPC-21, BioLegend) served as the isotype control for the anti-Ki-67-PE and anti-γ-H2AX-PE. The cells were washed twice in the permeabilisation buffer to remove the unbound antibodies and resuspended in the FACS buffer.

All washing and incubation steps were performed at room temperature. All samples were acquired on a BD LSRFortessa (BD Biosciences) and analysed using FlowJo (BD Biosciences).

## Flow cytometric detection of apoptotic cells

Apoptotic cells with externalised phosphatidylserine (PS) were detected using annexin V, a protein that binds to PS in a calcium-dependent way [57]. The cells were washed once with PBS, followed by a wash with annexin V binding buffer (BioLegend). The cells were then stained with 0.5 μg/ml annexin V-PE (BioLegend) and 1 μg/ml LIVE/DEAD fixable near-IR dye in annexin V binding buffer for 15 minutes. They were subsequently washed once with annexin v binding buffer containing 5% FCS and fixed with 4% formaldehyde diluted 1:1 with annexin V binding buffer. Finally, the cells were washed twice and immediately acquired on a BD LSRFortessa after resuspending in annexin V binding buffer containing 5% FCS. All washes and incubations were performed at room temperature. Data were analysed using FlowJo.

## ROS detection by flow cytometry

Flow cytometry-based ROS detection was performed using the CellROX Deep Red probe (ThermoFisher Scientific) that fluoresces upon ROS-dependent oxidation [23]. One hundred thousand cells from d2EGFP TBX4B and d2EGFP TBW 11.50 resuspended in complete growth medium were seeded into FACS tubes. One hundred micromolar tert-butyl hydroperoxide (TBHP), a ROS inducer [58] was used as the positive control and 250 μM N-acetyl-L-cysteine (NAC), a ROS scavenger [59] served as the negative control. The cells were cultured for 1 hour, and subsequently, 500 nM CellROX Deep Red probe was added and further incubated for 45 minutes. They were then washed once and resuspended in phenol red-free RPMI-1640 (ThermoFisher Scientific) containing 2% FCS. The cells were cultured for 15 minutes after adding 1 μM SYTOX blue dead cell labelling dye. The cells were immediately acquired on a BD FACSAria lll cell sorter (BD Biosciences). All incubations were performed in a humidified incubator at 37˚C, with 5% $CO_2$. The washing step was carried out at room temperature. FlowJo was used for data analysis.

## Flow cytometric analysis of cell-cycle progression

Cell-cycle progression was evaluated using a previously published [10] EdU pulse-chase assay using the Click-iT plus EdU AF-647 flow cytometry assay kit (ThermoFisher Scientific). Five hundred thousand cells from d2EGFP TBX4B and d2EGFP TBW 11.50 were cultured in complete growth medium in a 48-well plate (Corning). The cells were pulsed with 10 μM EdU for 1.5 hours. At the end of the 1.5-hr pulse the cells were washed three times in complete growth

medium to remove EdU that had not been taken up by the cells. Half of the cells were cultured in complete growth medium for further 6.5 hours–the chase period. The cells obtained after the EdU pulse were treated to the point of fixation and then stained alongside the cells collected at the end of the EdU chase. The samples collected at the end of EdU pulse and chase were washed once in PBS and stained for 5 minutes with 1 μg/ml LIVE/DEAD fixable near-IR dye. The cells were then washed once in FACS buffer and fixed with 4% formaldehyde for 30 minutes. Following two washes with FACS buffer, the cells were permeabilised with 0.1% Triton X-100 (ThermoFisher Scientific) for 15 minutes, followed by two more washes with FACS buffer. To detect EdU, 500 μl Click-iT plus reaction cocktail (438 μl PBS + 10 μl copper protectant + 2.5 μl AF-647 picolyl azide + 50 μl reaction buffer additive) was added to each sample resuspended in 100 μl FACS buffer and incubated for 30 minutes. The samples were washed twice in FACS buffer before resuspending in FACS buffer containing 1 μg/ml FxCycle Violet DNA dye (ThermoFisher Scientific). After 30 minutes, the samples were acquired on a BD LSRFortessa. The EdU pulse and chase were performed in a humidified 37˚C incubator with 5% $CO_2$. All staining and washing steps were carried out at room temperature.

The cell-cycle distribution of the EdU$^+$ cells–the population that is tracked over time was determined using the Dean-Jett-Fox algorithm in FlowJo. The percentage of EdU$^+$ cells that progressed to the G1 of the next cell-cycle (Figs 2E and S5) was determined using a previously described gating strategy [60].

## FACS sorting

All cell sorts were performed under sterile conditions on a BD FACSAria lll cell sorter housed in a containment level 3 (CL3) laboratory. All washes and incubations before cell sorting were performed at room temperature.

For sorting d2EGFP$^+$, d2EGFP$^-$ or viable cells, the cells were stained with 1 μg/ml LIVE/DEAD fixable near-IR dye for 5 minutes after washing once with PBS. They were then washed once and resuspended in RPMI-1640 without phenol red supplemented with 2% FCS before sorting viable d2EGFP$^+$, viable d2EGFP$^-$ or viable cells.

To sort viable apoptotic and non-apoptotic cells, the cells were washed once with PBS and then once with annexin V binding buffer. They were subsequently incubated for 15 minutes in annexin V binding buffer containing 0.5 μg/ml annexin V-PE and 1 μg/ml LIVE/DEAD fixable near-IR dye. Finally, the viable apoptotic and viable non-apoptotic cells were sorted following a wash and resuspension in annexin V binding buffer containing 2% FCS.

## Cycloheximide chase assay to determine d2EGFP half-life

One hundred thousand cells from clone d2EGFP TBW 11.50 were cultured complete growth medium in a 48-well plate. The cells were treated with either 10 μg/ml cycloheximide or DMSO, the vehicle control (both from Sigma-Aldrich) in duplicate. The cells harvested at 0-, 3-, and 6-hours post-treatment were stained with LIVE/DEAD fixable near-IR dye, fixed with 4% formaldehyde and acquired on a BD LSRFortessa as described above. The median fluorescence intensity (MFI) of viable d2EGFP$^+$ cells was determined using FlowJo. The natural logarithm (ln) of d2EGFP MFI was plotted against time, and linear regression analysis was performed using GraphPad Prism (GraphPad Software). Half-life (T$_{1/2}$) was calculated using the formula:

$$T_{1/2} = \frac{ln(2)}{\alpha}$$

Here, α is the slope of the fitted line, which represents the protein decay rate [61].

## Quantitative real-time reverse transcription PCR

RNA was extracted from unsorted, FACS-sorted d2EGFP$^+$ and d2EGFP$^-$ cells of d2EGFP TBX4B and d2EGFP TBW 11.50 clones using RNeasy plus mini kit (Qiagen). cDNA was synthesised from RNA using transcriptor first strand cDNA synthesis kit (Roche) with random hexamer primers. Each RNA sample included a no reverse transcriptase (RT) control to confirm the absence of genomic DNA. A master mix containing template cDNA, gene-specific primers and Fast SYBR Green master mix (ThermoFisher Scientific) was used on a ViiA 7 real-time PCR machine (ThermoFisher Scientific) for kinetic PCR amplification. LinRegPCR method [62] was used to determine the relative abundance of target mRNAs. The gene-specific values were normalised to their corresponding 18S rRNA values, which served as the internal PCR control. The gene-specific primers are listed in the S2 Table.

## Live-cell imaging and image analysis for quantifying Tax expression dynamics

Five thousand cells of d2EGFP TBX4B and d2EGFP TBW 11.50 resuspended in complete growth medium were seeded into a 96-well flat bottom plate (Corning) precoated with 1 mg/ml Poly-D-Lysine (Signa-Aldrich). Time-lapse live-cell imaging was performed using Incucyte S3 live-cell imaging system (Sartorius) housed in a humidified incubator kept at 37°C, 5% $CO_2$. Phase contrast and green fluorescence images were captured every 20 minutes for 30 hours using 20x objective, 300 ms acquisition time.

Image analysis and quantification were performed in ImageJ [63] unless otherwise stated. An ImageJ macro was used to process images by removing the background noise from green fluorescence images using the rolling ball algorithm and correcting the XY drift of the sequential images using the Correct 3D drift plugin. The phase contrast image stack was then exported and segmented to distinguish the cells from the background by 'training' the image set using the pixel classification method in ilastik [64]. The segmented phase stack was imported, converted to a binary stack, and merged with phase and green stacks to form a 3-channel hyper stack using an ImageJ macro. The Trackmate plugin [65] was used to detect and track the single cells of the binary channel whose mean green fluorescence intensity elevated above the background during the observation period to identify the different Tax expression patterns.

## Live-cell imaging and analysis for quantifying spontaneous and maximal reactivation of HTLV-1 plus-strand

Fifteen thousand FACS-sorted proviral silent (d2EGFP$^-$) cells from clones d2EGFP TBX4B and d2EGFP TBW 11.50 were resuspended in complete growth medium and seeded into a 96-well flat bottom plate precoated with 1 mg/ml Poly-D-Lysine. Protein kinase C (PKC) agonists bryostatin-1 (Apexbio) and prostratin (Abcam) were added at 10 nM and 300 nM, respectively. Fifty nanograms per millilitre phorbol-12-myristate-13-acetate (PMA) and 1 µM ionomycin (both from Abcam) were used as the positive control. DMSO was used as the vehicle control, while media alone was used to determine the frequency of cells enduring spontaneous reactivation. The concentrations of the latency-reversing agents (LRAs) and controls were selected based on Laird et al. [66]. Dead cells were labelled using 100 nM YOYO-3 Iodide dye (ThermoFisher Scientific). The 20x objective of the Incucyte S3 live-cell imaging system was used to acquire nine phase-contrast, green and red fluorescent images per well at a 4-hour frequency. The "Non-adherent Cell-by-Cell" image analysis module on Incucyte software was used to analyse the images using the parameters outlined in S3 Table to calculate the percentage of viable d2EGFP$^+$ cells.

## Live-cell imaging and image analysis for quantifying clumping

Five thousand FACS-sorted d2EGFP$^+$ and d2EGFP$^-$ cells from clones d2EGFP TBX4B and d2EGFP TBW 11.50 along with FACS-sorted viable cells from clones TCX 8.13 and TBW 13.50 were resuspended in complete growth medium and cultured in a 96-well flat bottom plate. For inhibition of LFA-1-ICAM-1 interaction or CCR4, five thousand FACS-sorted d2EGFP$^+$ cells from clone d2EGFP TBW 11.50 were pre-treated for one hour with inhibitors or controls at concentrations shown in Fig 4C. FACS-sorted d2EGFP$^-$ cells from the same clone were used as the negative control. One hundred nanomolar YOYO-3 Iodide dye was added to label dead cells. Whole-well phase contrast, green and red fluorescence images were acquired every 4 hours for 12 hours with 4x objective of Incucyte S3. The acquisition time for green and red channels was 300 ms and 400 ms, respectively.

An ImageJ macro was used to process images by removing the background noise from green and red fluorescence images using the rolling ball algorithm and eliminating the XY drift using the Correct 3D drift plugin. Phase contrast, green and red image stacks were exported to ilastik and segmented using the pixel classification workflow. The segmented stacks were imported and converted to binary stacks in ImageJ. The areas occupied by dead and d2EGFP$^-$ cells were removed from d2EGFP$^+$ binary stacks to calculate the mean object area of d2EGFP$^+$ cells. The mean object area of d2EGFP$^-$ cells was computed by excluding the regions comprised of d2EGFP$^+$ and dead cells from d2EGFP$^-$ binary stacks.

## Live-cell imaging and analysis for determining single-cell motility

d2EGFP$^+$ and d2EGFP$^-$ cells from clones d2EGFP TBX4B and d2EGFP TBW 11.50 were FACS-sorted, mixed 1:1 and seeded at 10000 cells per well in a 48 well plate with a pre-laid polydimethylsiloxane (PDMS) grid containing an array of 500 μm x 500 μm microwells (microsurfaces). Dead cells were labelled with 100 nM YOYO-3 Iodide. Phase contrast, green and red fluorescence images were captured at either 3- or 4-minute intervals for 1 hour using the 10x objective of Incucyte S3, with 300 ms and 400 ms acquisition time for green and red channels, respectively.

The background noise of fluorescence images was removed using the rolling ball algorithm, and the Correct 3D drift plugin was used to eliminate the XY drift of the successive images using an ImageJ macro. Phase contrast image stack was exported and segmented using the pixel classification workflow in ilastik. An ImageJ macro was used to import the segmented phase stack, convert it to a binary stack and merge it with phase contrast, green and red channels to form a 4-channel hyper stack. The detection and tracking of cells were performed using ImageJ's Trackmate plugin. The fluorescence intensity of the green channel was used to distinguish d2EGFP$^+$ and d2EGFP$^-$ cells. Single viable cells tracked for the entire observation period were included in the analysis. The mean speed and directionality were derived from the Trackmate plugin. The trajectory plots in Fig 5A were generated using the Chemotaxis and Migration Tool (ibidi). The mean square displacement of the individual cells was determined using a published MATLAB (MathWorks) script [67]. The motility patterns in Fig 5E were determined by fitting the log (MSD) vs log (t) by a straight line and calculating the gradient of the fits, where the goodness of the fit R$^2$ was greater than 0.8, as described previously [67]. Subdiffusive movement is when the upper bound of the 95% confidence interval of the gradient is below 1; diffusive movement is when the upper and lower bounds of the 95% confidence interval of the gradient are above and below 1, respectively; and superdiffusive movement is when the lower bound of the 95% confidence interval of the gradient is above 1.

### Estimation of Tax burst duration

A two-state random telegraph model [68] was used to estimate the average duration of Tax expression. Further details are given in the S1 Text.

### Cell counting

Eighteen microliters of cell suspension were mixed with 2 μl acridine orange-propidium iodide dye (Logos Biosystems), and 10 μl of the cell-dye mixture was loaded into a counting slide. Automated fluorescence cell counting was performed using LUNA-FL automated cell counter (Logos Biosystems).

### Statistical analysis

GraphPad Prism was used for plotting graphs and statistical analysis unless otherwise stated. The statistical tests used are specified in the figure legend.

## Supporting information

**S1 Fig. d2EGFP protein half-life.** Cells treated with either protein synthesis inhibitor–cyclo-heximide or vehicle control–DMSO were harvested after 0, 3 and 6 hours of treatment before fixation and flow cytometric acquisition. Natural logarithm-transformed MFI of d2EGFP$^+$ cells was calculated and plotted against time. Technical duplicates from a single experiment using clone d2EGFP TBW 11.50 are shown; the solid and dashed lines show the least-squares regression line and 95% confidence interval, respectively. d2EFP protein half-life was calculated as described in Materials and methods.
(PDF)

**S2 Fig. Spontaneous and maximal reactivation of HTLV-1 plus-strand.** FACS-sorted d2EGFP$^-$ cells from (A) d2EGFP TBX4B and (B) d2EGFP TBW 11.50 were cultured with PKC activators bryostatin-1 and prostratin and imaged every 4 hours for 20 hours. PMA and iono-mycin were used as the positive control for maximal reactivation, while media alone was used to determine the frequency of cells undergoing spontaneous HTLV-1 plus-strand reactivation. DMSO was used as the vehicle control. The percentage of viable d2EGFP$^+$ (Tax$^+$) cells at the end of the 20-hour culture. Data represent the mean and SEM from two independent experiments.
(PDF)

**S3 Fig. Comparable proportions of d2EGFP$^+$ (Tax$^+$) and d2EGFP$^-$(Tax$^-$) cells express ROS.** (A) Cells were either untreated or treated with a ROS inducer (THBP) or a ROS scavenger (NAC) for one hour. ROS-expressing cells were detected by flow cytometric analysis after labelling with CellROX Deep Red probe. ROS expression in cells under different treatment conditions. (B) ROS expression in d2EGFP$^+$ and d2EGFP$^-$ cells. Data represent the mean and SEM from two independent experiments using the clones d2EGFP TBX4B and d2EGFP TBW 11.50. Statistical analysis of panel A was performed using a one-way analysis of variance (ANOVA) followed by a Tukey's multiple comparisons test. An unpaired Student's t-test was used to analyse the data in panel B. $^{**}$ P $<$ 0.01, ns–not significant.
(PDF)

**S4 Fig. Cell-cycle phase distributions of total and EdU$^+$ cells in d2EGFP$^-$ (Tax$^-$) and d2EGFP$^+$ (Tax$^+$) cells at 1.5 and 8.0 hours.** Representative flow cytometric plots of clone d2EGFP TBX4B.
(PDF)

**S5 Fig. Gating strategy used to determine the cells among EdU$^+$ cells of d2EGFP$^-$ (Tax$^-$) and d2EGFP$^+$ (Tax$^+$) cells that had progressed to the G1 phase of the next cell-cycle after 8 hours.** Representative flow cytometric plots of clone d2EGFP TBX4B.
(PDF)

**S6 Fig. The fraction of EdU$^+$ cells among cells that express Tax is lower.** The percentage of cells among d2EGFP$^+$ (Tax$^+$) and d2EGFP$^-$ (Tax$^-$) cells that had taken up EdU at the end of the 1.5-hour pulse. Data are mean and SEM from two independent experiments using clones d2EGFP TBX4B and d2EGFP TBW 11.50. Statistical analysis was performed using an unpaired Student's t-test. $^{**}$ P < 0.01.
(PDF)

**S7 Fig. Cell clump formation in Tax-expressing cells.** The mean object area of d2EGFP$^+$ (Tax$^+$) and d2EGFP$^-$ (Tax$^-$) cells of clones d2EGFP TBX4B and d2EGFP TBW 11.50 and viable cells of clones TCX 8.13 and TBW 13.50 at the beginning (0 h) and end (12 h) of live-cell imaging. The data depict the mean and SEM from two independent experiments.
(PDF)

**S8 Fig. Higher percentage of d2EGFP$^+$ (Tax$^+$) cells express ICAM-1 and CCR4.** The proportion of d2EGFP$^+$ (Tax$^+$) and d2EGFP$^-$ (Tax$^-$) cells that express ICAM-1 or CCR4 was assessed by flow cytometric analysis. Data are mean and SEM from two independent experiments using clones d2EGFP TBX4B and d2EGFP TBW 11.50. An unpaired Student's t-test was used for statistical analysis. $^{**}$ P < 0.01.
(PDF)

**S9 Fig. Tax-expressing cells produce increased levels of *CCL22* transcripts.** (A) The expression levels of *tax*, (B) *d2EGFP*, (C) *CCL22*, and (D) *sHBZ* transcripts among unsorted, FACS-sorted d2EGFP$^+$ (Tax$^+$) and d2EGFP$^-$ (Tax$^-$) cells were determined using RT-qPCR as described in Materials and methods. The data depict the mean and SEM of two PCR technical replicates of clones d2EGFP TBX4B and d2EGFP TBW 11.50 from a single experiment.
(PDF)

**S1 Table. Details of T-cell clones used in this study.**
(DOCX)

**S2 Table. Gene-specific primers used for RT-qPCR.**
(DOCX)

**S3 Table. Image capturing and analysis parameters used to quantify the frequency of cells undergoing spontaneous and maximal HTLV-1 plus-strand reactivation.**
(DOCX)

**S1 Video. Most Tax-expressing cells maintain continuous expression.** Representative video of clone d2EGFP TBX4B. The scale bar is 20 μm.
(AVI)

**S2 Video. Tax expression is associated with reduced motility.** Representative video of clone d2EGFP TBX4B. The scale bar is 20 μm.
(AVI)

**S1 Text. Estimation of Tax burst duration using a two-state random telegraph model.**
(DOCX)

**S1 Data. Excel workbook containing the numerical data used to generate the figures.** Each named sheet within the workbook contains the numerical values of the corresponding figure panel.
(XLSX)

## Acknowledgments

We thank Mrs Parisa Amjadi of the CL3 Cell Sorting Facility at the Center for Immunology and Vaccinology of Imperial College London for cell sorting; Professor Chou-Zen Giam of Uniformed Services University for supplying the SMPU-18x21-EGFP plasmid; and Professor Teruna Siahaan from the University of Kansas for providing cLAB.L and RD-LBEC cyclic peptides.

## Author Contributions

**Conceptualization:** Saumya Ramanayake, Charles R. M. Bangham.

**Data curation:** Saumya Ramanayake.

**Formal analysis:** Saumya Ramanayake, Abhyudai Singh.

**Funding acquisition:** Charles R. M. Bangham.

**Investigation:** Saumya Ramanayake.

**Methodology:** Saumya Ramanayake, Dale A. Moulding.

**Project administration:** Charles R. M. Bangham.

**Resources:** Yuetsu Tanaka, Charles R. M. Bangham.

**Software:** Saumya Ramanayake.

**Supervision:** Charles R. M. Bangham.

**Validation:** Saumya Ramanayake, Abhyudai Singh, Charles R. M. Bangham.

**Visualization:** Saumya Ramanayake.

**Writing – original draft:** Saumya Ramanayake, Charles R. M. Bangham.

**Writing – review & editing:** Saumya Ramanayake, Dale A. Moulding, Yuetsu Tanaka, Abhyudai Singh, Charles R. M. Bangham.

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
