## [Decision Letter · Decision Letter 0]

7 Sep 2022

Dear Prof. Bangham,

Thank you very much for submitting your manuscript "Dynamics and consequences of the HTLV-1 proviral plus-strand burst" for consideration at PLOS Pathogens. As with all papers reviewed by the journal, your manuscript was reviewed by members of the editorial board and by three independent reviewers. In light of the reviews (below this email), we would like to invite the resubmission of a significantly-revised version that takes into account the reviewers' comments.

In particular, there were 2 major findings that the reviewers found exciting that needed experimental evidence regarding mechanism. This includes a better understanding of the cause of cell aggregation (and loss of motility) in Tax-expressing cells and how whether (and how) loss of Tax expression in previously expressing cells resulted in a competitive replication advantage compared to cells that never expressed Tax.

We cannot make any decision about publication until we have seen the revised manuscript and your response to the reviewers' comments. Your revised manuscript is also likely to be sent to reviewers for further evaluation.

Sincerely,

Susan R. Ross, PhD

Section Editor

PLOS Pathogens

Susan Ross

Section Editor

PLOS Pathogens

Kasturi Haldar

Editor-in-Chief

PLOS Pathogens

orcid.org/0000-0001-5065-158X

Michael Malim

Editor-in-Chief

PLOS Pathogens

orcid.org/0000-0002-7699-2064

Reviewer's Responses to Questions

**Part I - Summary**

Reviewer #1: In this manuscript, the authors examine the dynamics of Tax expression and functional consequences of plus-strand bursts in two HTLV-1-infected T cell clones (TBX4B and TBW 11.50) expressing a short half-life eGFP Tax reporter system. The authors performed live-cell imaging of the Tax-expressing cells, which revealed different patterns of Tax expression with both clones mainly showing continuous expression of Tax. In the short-term, Tax-expressing cells exhibited decreased proliferation, increased DNA damage, slower progression through G2/M of the cell cycle and increased apoptosis. In a longer duration experiment of 14 days, Tax expression progressively diminished which coincided with increased cell proliferation, fewer apoptotic cells and a greater net expansion of the Tax+ population. Finally, Tax-expressing cells formed cell clumps and had reduced motility at the single-cell level.

This is an important study for the field since it uses a physiologically relevant model system to examine the dynamics of Tax expression and the functional consequences of the plus-strand burst. However, some of the functional effects attributed to Tax expression in this manuscript are not congruent with many published studies from over the last 20+ years. This could potentially be due to different model systems and/or effects of Tax on other viral genes expressed from the plus-strand or on HBZ. Therefore, the authors should exert caution with regard to data interpretation and formulation of conclusions since some of the functional outcomes may not be directly attributable to Tax, but rather a cumulative effect of the plus-strand bursts. Other points below should be addressed/clarified by the authors.

Reviewer #2: In this manuscript, Ramanayake et al. characterize the kinetics and outcomes of viral reactivation in two human T cell clones latently infected by HTLV-1. These two cell lines were stably transduced with a reporter construct that expressed a short-lived destabilized form of the enhanced green fluorescent protein (dsGFP), whose expression is driven by the viral trans-activator, Tax. This system allows viral replication and reactivation to be analyzed at the single cell level. The authors report results to indicate that Tax expression correlated with the induction of DNA double-strand breaks, cell cycle cessation, and apoptosis initially. Later, resumption of cell proliferation occurred within a given population of cells undergoing reactivation and correlated with decreased or loss of Tax expression. Finally, cells undergoing viral reactivation showed a propensity to aggregate and were less motile.

The study confirmed the relevance of earlier results done in lymphoid and non-lymphoid cell lines that showed the importance of Tax-associated DNA damage on the induction of apoptosis or senescence/cell cycle arrest. Further, the induction of ROS by Tax reported previously was not observed in T cells undergoing latency reactivation. The duration of Tax expression after reactivation in both cell lines is also longer (94 and 417 hours vs the 19 hours reported in ref. 10). Overall, the study was performed meticulously. The results recapitulated what had been previously reported, especially in Mahgoub et al. (cited in ref. 10), using the identical approach, albeit with additional details.

Reviewer #3: The group co-discovered the burst expression of the plus-strand of HTLV-1 some years ago and have been studying the relevance of this phenomenon. In this paper, they combined imaging-based single cell study with mathematical modeling to present new interpretation/paradigm of the finding. Curiously, when the cells express Tax, it acts against proliferation, prevention of apoptosis, or subsidization of the DNA damage response. However, when these cells “lost” Tax-expression, they seem to gain advantages over cells that have never experienced Tax expression for the speed of cell division and consequent expansion of those clones, which presents another support for the hypothesis that Tax contributes to the oncogenesis of HTLV-1.

**Part II – Major Issues: Key Experiments Required for Acceptance**

Reviewer #1: 1) The functional consequences of Tax expression in the short-term in this study are surprising given the extensive literature on Tax. The authors found that Tax expression is associated with decreased cell proliferation, slower cell-cycle progression and increased apoptosis- attributes not normally associated with a viral oncogene. Although different model systems were used, many studies have clearly established that Tax is anti-apoptotic. A recent paper (Mahgoub et al. 2018, PNAS, 115: e1269-1278) showed that transient expression of Tax induces anti-apoptotic machinery at the single-cell level. The current study also showed no effect of Tax on ROS production in cells although two previous studies have shown that Tax can increase ROS production (Takahashi et al. 2013, Blood; and Kinjo et al. 2010, J. Virol). How can the authors explain the discordance with these published studies? As described above, Tax regulation of plus-strand viral regulatory genes (or HBZ) is likely to influence certain cellular functional outcomes. Therefore, how can the authors disentangle direct effects of Tax versus indirect effects (i.e., other viral genes)?

2) It has been long known that Tax expression promotes cell adhesion/clumping (Takahashi et al. 2002 Virology, 302: 132-43). However, the authors failed to identify the adhesion molecules important for the clumping, which appear to be independent of ICAM1-LFA1 and CCL22-CCR4 interactions. How can the authors be certain that the inhibitors and treatments used in Fig. 4C functioned as intended in the absence of positive controls?

Reviewer #2: 1. In Fig. 1, it would be useful to know the fraction of T cells in each of the two latently infected clones that undergo spontaneous reactivation (i.e., Tax+). Does the clone that expresses Tax longer (TBX4B) contain more cells undergoing spontaneous reactivation? What fraction of cells are “fluctuating” in Tax expression, i.e., experiencing repeated cycles of reactivation and latency? Lengthening the time of video recording may provide some clues.

2. The decline in Tax+ cells within one week after reactivation could be a result of cell cycle arrest, apoptosis, or senescence in response to the DNA damage caused by Tax. Tax is known to dramatically induce the expression and stabilization of CDK inhibitors, p21 and p27. The former is intimately associated with senescence induction. The authors should consider staining the Tax+ cells for nuclear p21 and track the fate of such cells for a longer duration to assess whether cell cycle arrest and senescence contribute to the decline in Tax+ cells and whether these cells recover from the arrest. Earlier publications in this regard should also be cited and discussed.

3. The cell clumping observed could be due to the expression of Env on the cell surface and Env interaction with GLUT1 or other cell surface receptors. The author should include this experiment in the analysis.

Reviewer #3: While this paper presents novel findings, it does not seem to provide drastic information that would advance the field substantially. The progress seems rather incremental and it is descriptive than analytical. Some major findings are presented without reasonable explanation.

For example, they did studies on cell adhesion, but did not come up with positive results. LFA-1 and CCR4 were excluded from candidates, but it was not conclusive and seems to remain open-ended. This part lacks convincing impression. Would the transcriptome analysis provide any plausible new candidates on this (may not want to show the detail of the classified information, but implying sentences can be added to the text).

Tax-expression renders the cells that expressed Tax advantageous to non-expressing cells. This is an interesting observation, in agreement with previous observations that HTLV-1+ cells can maintain Tax-interactome in the absence of Tax, in strong support that Tax is a prime factor for HTLV-1’s leukemogensis. However, there is no underlying mechanism shown in this work. This is surprising knowing the high quality of the works usually demonstrated by this group. For example, it is attractive to hypothesize that the cells once expressed Tax seem to remember it by maintaining an environment (transcriptional, epigenetic, or else?). Whether or not cells that once expressed Tax would maintain tax-interactome can be shown relatively easily and the addition of such information would enhance the relevance of this work.

**Part III – Minor Issues: Editorial and Data Presentation Modifications**

Reviewer #1: 1) The authors state in lines 152-154 that the average duration of Tax expression as estimated by mathematical modeling is 94 hours and 417 hours in the two clones. If this is true why does Tax expression progressively decline between 0-14 days (Fig. 3B)?

2) Statistical analysis is needed for data points in Fig. 3B-E. There doesn’t appear to be appreciable differences in Annexin V+ cells at days 11 and 14 (Fig. 3C).

3) Line 259 should read: “Tax-expressing cells form cell clumps”

Reviewer #2: 1. The two HTLV-1-infected T cell clones fit the description of HTLV-1 immortalized T cells, i.e., continuous proliferation and IL-2-dependent growth. The authors should provide more details about these clones.

2. In video S1, the time-lapse video was for 30 hours. During this period, many of the Tax+ cells remained as single cells. They appeared to increase in size and attempted to divide but failed to complete cytokinesis, reminiscent of Tax-expressing cells that eventually become senescent (PMID: 21209109). The authors should discuss the video in this context.

3. The Mahgoub et al. paper (PMID: 29358408) showed that sporadic Tax expression during reactivation leads to the expression of survival factors that facilitates the expansion of other T cells in an altruistic manner. This earlier observation by the Masao group could explain the results in Fig. 3E and answer the question of the mechanism underlying the post-Tax burst proliferation surge, and should be mentioned, cited, and discussed.

4. An issue not addressed in the study is whether cells undergoing reactivation can infect naïve T cells via the virological synapse. Comments or experiments in this regard would be valuable.

5. The DNA double-strand breaks and senescence induced by Tax are caused by NF-kappa B hyperactivation and are associated with the accumulation of R-loops. Appropriate literature in this regard should be cited.

Reviewer #3: (No Response)

PLOS authors have the option to publish the peer review history of their article (what does this mean?). If published, this will include your full peer review and any attached files.

Reviewer #1: No

Reviewer #2: No

Reviewer #3: No
---

## [Decision Letter · Decision Letter 1]

15 Nov 2022

Dear Prof. Bangham,

We are pleased to inform you that your manuscript 'Dynamics and consequences of the HTLV-1 proviral plus-strand burst' has been provisionally accepted for publication in PLOS Pathogens.

Best regards,

Susan R. Ross, PhD

Section Editor

PLOS Pathogens

Susan Ross

Section Editor

PLOS Pathogens

Kasturi Haldar

Editor-in-Chief

PLOS Pathogens

orcid.org/0000-0001-5065-158X

Michael Malim

Editor-in-Chief

PLOS Pathogens

orcid.org/0000-0002-7699-2064

Reviewer Comments (if any, and for reference):

Reviewer's Responses to Questions

**Part I - Summary**

Reviewer #1: The revised manuscript is improved and my previous concerns were mostly addressed by the authors. With regard to Figure 4C, the authors cannot say with certainty that the ICAM-1 inhibitors actually worked in their experiment since these are negative results and there were no positive controls. The authors now mention a parallel study that provides evidence of the PLC gamma pathway and IL-2 in clump formation, however whether ICAM-1 is involved in clump formation remains unclear.

Reviewer #2: The authors have substantively addressed most of the comments raised in the previous critique. Additional discussion to clarify the following point will greatly help the readers.

The two HAM/TSP-patient-derived clones used in the study were apparently “selected” both in vivo and in vitro to survive HTLV-1 infection and reactivation. They may contain adaptive changes that mitigate the cytotoxic effects of the initial viral infection and facilitate cell survival under sustained Tax expression. As such, the experimental system approximates natural viral reactivation better, but it is still somewhat removed from primary cells newly infected by HTLV-1.

Reviewer #3: The group co-discovered the burst expression of the plus-strand of HTLV-1 some years ago and have been studying the relevance of this phenomenon. Since the prevailing paradigm before the discovery was that Tax-1 that is encoded by the plus-strand rarely gets transcribed in established leukemic ATL cells in patients. This casts a conundrum as to the oncogenic role of Tax-1 in the development of ATL. Due to the studies two groups including this one, the paradigm has been rewritten.

In this paper, they combined imaging-based single cell study with mathematical modeling to present new interpretation/paradigm of the finding. The use of these technologies/strategy is commendable. Curiously, when the cells express Tax, it acts against proliferation, prevention of apoptosis, or subsidization of the DNA damage response. However, when these cells “lost” Tax-expression, they seem to gain advantages over cells that have never experienced Tax expression for the speed of cell division and consequent expansion of those clones, which presents another support for the hypothesis that Tax significantly contributes to the oncogenesis of HTLV-1.

The only reservation to this paper - while this paper presents novel and interesting findings some of which confront with previous findings, it does not seem to represent a dramatic advancement. The progress seems rather incremental, and it is descriptive than analytical. However, this publication should provoke the field to reproduce, or test alternative hypotheses and hence could bring long-term major advancement.

**Part II – Major Issues: Key Experiments Required for Acceptance**

Reviewer #1: N/A

Reviewer #2: none

Reviewer #3: None - they have been solved with the current revision

**Part III – Minor Issues: Editorial and Data Presentation Modifications**

Reviewer #1: N/A

Reviewer #2: none

Reviewer #3: None- the authors addressed minor issues

PLOS authors have the option to publish the peer review history of their article (what does this mean?). If published, this will include your full peer review and any attached files.

Reviewer #1: No

Reviewer #2: No

Reviewer #3: No

---

## [Editor Report · Acceptance letter]

21 Nov 2022

Dear Prof. Bangham,

We are delighted to inform you that your manuscript, "Dynamics and consequences of the HTLV-1 proviral plus-strand burst," has been formally accepted for publication in PLOS Pathogens.

Best regards,

Kasturi Haldar

Editor-in-Chief

PLOS Pathogens

orcid.org/0000-0001-5065-158X

Michael Malim

Editor-in-Chief

PLOS Pathogens

orcid.org/0000-0002-7699-2064